# The acute myeloid leukemia associated AML1-ETO fusion protein alters the transcriptome and cellular progression in a single-oncogene expressing in vitro induced pluripotent stem cell based granulocyte differentiation model

Esther Tijchon[1], Guoqiang Yi[1], Amit Mandoli[1], Jos G. A. Smits[1], Francesco Ferrari[1], Branco M. H. Heuts[1], Falco Wijnen[1], Bowon Kim[1], Eva M. Janssen-Megens[1], Jan Jacob Schuringa[2], Joost H. A. Martens[1] *

1 Radboud University, Department of Molecular Biology, Faculty of Science, Nijmegen Centre for Molecular Life Sciences, Nijmegen, the Netherlands, 2 Department of Hematology, University Medical Centre Groningen, Groningen, The Netherlands

* j.martens@ncmls.ru.nl

## Abstract

Acute myeloid leukemia (AML) is characterized by recurrent mutations that affect normal hematopoiesis. The analysis of human AMLs has mostly been performed using end-point materials, such as cell lines and patient derived AMLs that also carry additional contributing mutations. The molecular effects of a single oncogenic hit, such as expression of the AML associated oncoprotein AML1-ETO on hematopoietic development and transformation into a (pre-) leukemic state still needs further investigation. Here we describe the development and characterization of an induced pluripotent stem cell (iPSC) system that allows in vitro differentiation towards different mature myeloid cell types such as monocytes and granulocytes. During in vitro differentiation we expressed the AML1-ETO fusion protein and examined the effects of the oncoprotein on differentiation and the underlying alterations in the gene program at 8 different time points. Our analysis revealed that AML1-ETO as a single oncogenic hit in a non-mutated background blocks granulocytic differentiation, deregulates the gene program via altering the acetylome of the differentiating granulocytic cells, and induces t(8;21) AML associated leukemic characteristics. Together, these results reveal that inducible oncogene expression during in vitro differentiation of iPS cells provides a valuable platform for analysis of aberrant regulation in disease.

## Introduction

Acute myeloid leukemia (AML) is an aggressive heterogeneous disease that is characterized by chromosomal translocations, insertions/deletions and point mutations. The most common chromosomal translocation in AML (10% of total AML) is t(8;21), which generates the

**Data Availability Statement:** Raw data files for the RNA and ChIP sequencing and analysis have been deposited in the GEO database under accession number: GSE119901. All other relevant data are within the paper and its Supporting Information files.

**Funding:** This work was funded by KWF (project # KUN2011-4937) and KIKA (project 311).

**Competing interests:** The authors have declared that no competing interests exist.

AML1-ETO oncofusion protein and is mostly associated with a favorable prognosis [1, 2] The t(8;21) translocation is considered to be the initiating (driver) hit for further AML development and can sometimes already be observed *in utero* [3].

In mice, secondary mutations are required to develop a full-fledged leukemia in the presence of intact AML1-ETO protein, although it was found that expression of a truncated AML1-ETO9a protein can give rise to full-blown leukemias [4–6]. In humans, t(8;21) AMLs are characterized by cooperating genetic aberrations, such as mutations of growth factor receptors, proto-oncogenes, and transcription factors such as stem cell factor receptor (c-Kit), FMS-related tyrosine kinase (FLT3), NRAS, PU.1 and AML1 [7]. This oncogenic cooperation results in enhanced self-renewal and differentiation arrest in hematopoietic progenitor and myelomonocytic cells [8, 9].

AML1 (RUNX1) functions as a DNA-binding transcription factor that is essential for fetal and adult hematopoiesis, and forms a complex with the core-binding factor β (CBFβ) [10, 11]. Eight-twenty-one (ETO) functions as a corepressor by recruiting the NCoR/SMRT/HDAC complexes [12, 13]. AML1-ETO consists of the DNA-binding domain (RUNT) of AML1 and four conserved domains of ETO. AML1-ETO was defined as a transcriptional repressor, although this does not represent all its biological functions [2, 14]. Over the last years it has become clear that the AML1-ETO oncofusion protein can also activate transcription by a mechanism involving p300 interactions [15]. Recently, it was reported that AML1-ETO induces self-renewal by the induction of C/D box snoRNA/RNPs, whereas another study defined FOXO1 as a critical regulator of the self-renewal program in AML1-ETO preleukemia cells [16, 17]. However, the exact molecular and biological mechanisms by which AML1-ETO initiates leukemia remain elusive. This is partly due to the frequent usage of mouse model systems, which do not fully recapitulate the human disease, or the usage of human cell lines and primary human AML cells, which harbor additional genomic aberrations. To study the specific role of AML1-ETO during human hematopoiesis additional model systems are required.

Recently, we have shown that leukemia maintenance is dependent on a balance between AML1-ETO, RUNX1, and ERG expression, and that disturbed RUNX1 or ERG function results in AML1-ETO overdose and cell death [18]. Using human induced pluripotent stem cells (iPSCs) that inducible express AML1-ETO we confirmed that increased AML1-ETO expression resulted in reduced cell viability. To identify the molecular mechanisms by which AML1-ETO initiates AML, we further exploited this human iPSC system expressing the AML1-ETO oncofusion gene by performing myeloid differentiation followed by global transcriptome and epigenome analysis. Our results reveal that AML1-ETO alters the myeloid gene program to primarily inhibit granulocyte development by affecting signaling and differentiation pathways.

## Experimental procedures

### Cell culture

An inducible AML1-ETO iPS cell line was generated as described previously by Mandoli et al. [18]. The iPSCs were cultured in E8 medium (Life Technologies) supplemented with 1% Penicillin/Streptomycin on vitronectin coated plates at 37˚C. Cells were split as colonies every 3–4 days on fresh vitronectin coated plates using ReLeSR (Stemcell Technologies). Kasumi-1 (DSMZ) was routinely cultured in RPMI 1640 supplemented with 10% FCS and 1% pen/strep at 37 ˚C.

### Myeloid differentiation

For myeloid differentiation the AML1-ETO iPSCs were dissociated using TrypLE and resuspended in E8 medium supplemented with 1% Penicillin/Streptomycin and 10 μg/ml Rock

inhibitor. Cells were seeded at a density of 150 cells/well on geltrex-coated 6 well plates, so that only four to five colonies appear in each well. The cells were maintained in E8 medium till the colonies reach a size of 500μM. At day 0 of differentiation, the E8 medium was replaced by stemline II medium supplemented with 1% Penicillin/Streptomycin, 1:100 insulin-transferrin-selenium-ethanolamine (ITS-X) and cytokines (20 ng/ml BMP4, 40 ng/ml VEGF, 50 ng/ml bFGF). After 3 days the medium was refreshed and on day 6 the cytokines were replaced with a specific cytokine cocktail to induce monocyte (50 ng/ml SCF, 50 ng/ml FLT3-L, 50 ng/ml IL-3, 50 ng/ml M-CSF and 10 ng/ml TPO) and granulocyte (50 ng/ml SCF, 50 ng/ml IL-3, 50 ng/ml G-CSF, 5 ng/ml TPO) differentiation. The medium was refreshed every 3–4 days and 14 ng/ml doxycycline was added to induce AML1-ETO expression at day 0, 6, 10 or 14 of differentiation. The AML1-ETO expressing cells were kept continuously in doxycycline upon further analysis.

## Flow cytometric analysis

During *in vitro* differentiation myeloid cells were produced in the supernatant and collected at different time points for flow cytometric analysis. Cells were washed in PBA buffer (1% BSA in PBS) and pre-incubated in 2% human serum to inhibit unspecific antibody binding. The cells were stained with a progenitor (CD34-CD45), monocyte (CD14-CD33-CD45-CD16) or neutrophil (CD15-CD45-CD16) antibody cocktail. After staining the cells were fixed in 1% paraformaldehyde (PFA) and analyzed on the FACS Calibur within 3 days. The data was collected and analyzed by the FlowJo software. T-testing was used to validate significant changes in cell populations.

## Cytospin

For morphological analysis, $5x10^4$ iMonocyte (iPSC derived Monocytes) or iGranulocyte suspensions were collected and spotted on a glass slide by centrifugation for 10 minutes at 800xg and air-dried at room temperature for at least one hour. Cells were fixed and stained with May-Grünwald and Giemsa staining after which the cells were washed and mounted using permount solution. Images were made on a VisionTek microscope using the 20x objective.

## RNA-seq

iMonocytes and iGranulocytes were purified by MACS sort using CD14 and CD16 beads (Miltenyi Biotec). Total RNA was isolated from $1x10^6$ purified CD14$^+$ iMonocytes and CD16$^+$ iGranulocytes as well as from control and AML1-ETO expressing iMonocytes and iGranulocytes using the RNeasy mini kit (Qiagen) and on-column DNaseI treatment. Ribosomal RNA was depleted from 500ng of RNA using the ribo-zero rRNA removal kit according manufacturer's protocol (Illumina). RNA was fragmented in 300 bp fragments by incubation in 5x fragmentation buffer (200 mM Tris-acetate, 500 mM Potassium Acetate, 150 mM Magnesium Acetate (pH 8.2)) for 7.5 minutes at 94˚C. First strand cDNA synthesis was performed using superscript III (Life Technologies), followed by synthesis of the second cDNA strand. Libraries were generated using the Kapa hyper prep kit (KAPA Biosystems).

## ChIP-seq and ChIPmentation

**ChIPmentation.** iMonocytes and igranulocytes were purified by MACS sort using CD14 and CD16 beads (Miltenyi Biotec). The purified cells were crosslinked with 1% formaldehyde for 10 minutes at room temperature at a concentration of $15x10^6$ cells/ml. The fixed cells were sonicated for 5 minutes (30 sec on; 30 sec off) using the Diagenode Bioruptor and centrifuged

at maximum speed for 10 minutes. Chromatin of $2x10^5$ cells was incubated overnight in dilution buffer (167mM NaCl, 16,7 mM Tris (pH 8), 1,2mM EDTA, 1% Triton X-100) with 1 μg H3K4me3 antibody at 4˚C. ProtA/G beads were blocked with dilution buffer (+0,15% SDS) and incubated with the chromatin-Ab for one hour at 4˚C. Beads were washed with three different wash buffers and the DNA library was prepared using the Nextera DNA sample prep kit (Illumina) described by Schmidl et al. [19].

**ChIP-seq.** Control and AML1-ETO iMonocyte and iGranulocyte suspensions were cross-linked with 1% formaldehyde for 10 minutes at room temperature at a concentration of $15x10^6$ cells/ml. The fixed cells were sonicated for 4 minutes (30 sec on; 30 sec off) using the Diagnode Bioruptor and centrifuged at maximum speed for 10 minutes. Chromatin of about $5x10^5$ cells was incubated overnight in dilution buffer (167mM NaCl, 16,7 mM Tris (pH 8), 1,2mM EDTA, 1% Triton X-100) with 1 μg H3K27ac antibody at 4˚C. ProtA/G beads were blocked and incubated with the chromatin-Ab for one hour at 4˚C. Beads were washed with three different wash buffers and chromatin was eluted from the beads. DNA-proteins were de-crosslinked (200mM NaCl and 4 μl proteinase K (10mg/ml)), by incubation for four hours at 65˚C and samples were purified using the Qiaquick MinElute PCR purification kit according manufacturer's protocol. The procedure was performed according the Blueprint Histone ChIP protocol (http://www.blueprint-epigenome.eu/UserFiles/File/Protocols/Histone_ChIP_July2014.pdf).

## Illumina high-throughput sequencing

Libraries were generated using the Kapa hyper prep kit. End repair and A-tailing was performed on the double strand DNA using end repair and A-tailing buffer. Subsequently, the adapters were ligated and a post-ligation cleanup was performed using Agencourt AMPure XP reagent. The libraries were amplified by PCR using the Kapa Hifi hotstart readymix and primer mix for 10 cycles. Samples were purified using the Qiaquick MinElute PCR purification kit and 300 bp fragments were selected from E-gel and the size was checked on the agilent bioanalyzer. Samples were sequenced on the Illumina HiSeq 2000, generating 42 bp single end reads. The sequencing reads were mapped to the human genome assembly hg19 using Burrows-Wheeler Alignment Tool (BWA) for ChIP-Seq or Tophat2 (bowtie) for RNA-Seq. For each base pair in the genome, the number of overlapping sequence reads was determined, averaged over a 10bp window and visualized in the UCSC genome browser (http://genome.ucsc.edu).

## Bioinformatic analysis

### ChIP-seq data

Sequencing reads were aligned to the human genome hg19 using bwa. For peak calling MACS2 [20] was used to call the peaks. To find different expression patterns of H3K27ac deeptools analysis was performed by normalizing the tagcounts using RPKM and plotting heatmaps. Peak annotation was performed using HOMER annotatepeaks.pl to define the TSS, exon, intron and intergenic regions (http://homer.ucsd.edu/homer/ngs/annotation.html).

The AML1-ETO specific motifs were identified using FIMO motif scanning from the MEME-suite.org/tools website.

### RNA-seq data

RNA-Seq reads were mapped to the human reference genome hg19 and subsequently used for bioinformatics analysis. The RPKM values for RefSeq genes were computed to analyze the

expression levels. Raw tag counts were normalized using the DESeq2 package in R and then pairwise comparisons were performed (2-fold change, p-value <0.05) to determine the differentially expressed genes in each condition. To find different expression patterns K-means clustering was applied to the normalized dataset described above.

### Accession numbers

Raw data files for the RNA and ChIP sequencing and analysis have been deposited in the GEO database under accession number: GSE119901.

## Results

### Hematopoietic differentiation of iPSCs

Most molecular studies on the AML associated oncogene AML1-ETO in human reported the function either in cell lines or patients blast cells that harbor additional mutations and represent end stage leukemia. To investigate the molecular mechanisms underlying AML1-ETO induced leukemogenesis in human without the context of additional mutations, we generated a human induced pluripotent stem cell model that allows expression of AML1-ETO via a doxycycline (dox) inducible promoter during *in vitro* hematopoietic differentiation [18]. For this we used a recently established iPSC clone derived from megakaryoblasts [20]. First, based on previous analysis [21, 22] we established protocols for differentiation of induced pluripotent stem cells towards the monocytic and granulocytic lineage (Fig 1A) and probed cell morphology and cell surface marker expression. Flow cytometry analysis of monocyte lineage differentiated cells revealed a cell population enriched for CD14$^+$ and CD16$^+$ cells. CD14 is a marker for monocytes and macrophages, while the combination with CD16 is used to distinguish classical (CD14$^+$) from non-classical (CD14$^+$/CD16$^+$) monocytes. Hence, we named these iPSC derived cell population iMonocytes/iMacrophages. Similarly, for granulocyte lineage differentiation a cell population dominated by CD15$^+$CD16$^+$ markers was identified, which we named iGranulocytes (Fig 1B). Independent replicates of the experiment revealed reproducible results (S1 Fig), validating the robustness of the differentiation protocols. To verify the cell surface marker analysis, the morphology of differentiated cells was visualized by cytospin using May-Grünwald and Giemsa staining. This confirmed the presence of cells that morphologically resemble monocytes and granulocytes in the two differentiation protocols (Fig 1C), showing differentiation of iPSCs towards these myeloid lineages. Next we isolated RNA from the iMonocyte and iGranulocyte cells and performed RNA-seq. We included non-differentiated iPSC as well as a cell population obtained during early differentiation and enriched for CD45 in our analysis. Principal component analysis (PCA) of the transcriptome showed that PC1 could separate the undifferentiated iPSCs from differentiated hematopoietic cells, including iMonocytes/iMacrophages (iCD14) and iGranulocytes (iCD16) (Fig 1D). In contrast, PC2 separated iCD14 from iCD16 cells, displaying distinct myeloid lineages. Subsequent K-means clustering revealed clustering between the differentiated cells, corroborating the PCA results (Fig 1E). In addition, it allowed identification of five different gene signatures, with cluster 1 (green) harbouring genes specifically expressed in iPSCs such as POU5F1, DNMT3B, SOX2, DPPA4, LIN28, FGFR4, and ZFP42, which all represent previously well characterized stem cell genes [23]. In contrast, in cluster 2 (yellow) genes specifically expressed in iGranulocytes such as VNN1, CR1, MGAM and FCGR3B were identified, while in cluster 3 (purple) genes specific for hematopoietic cells such as SPI1 were found. Cluster 4 (red) genes were mainly repressed in iMonocytes/iMacrophages and included MMP8, CTSG, ANPEP, GATA1, while cluster 5 (blue) genes were upregulated in iMonocytes and iGranulocytes.

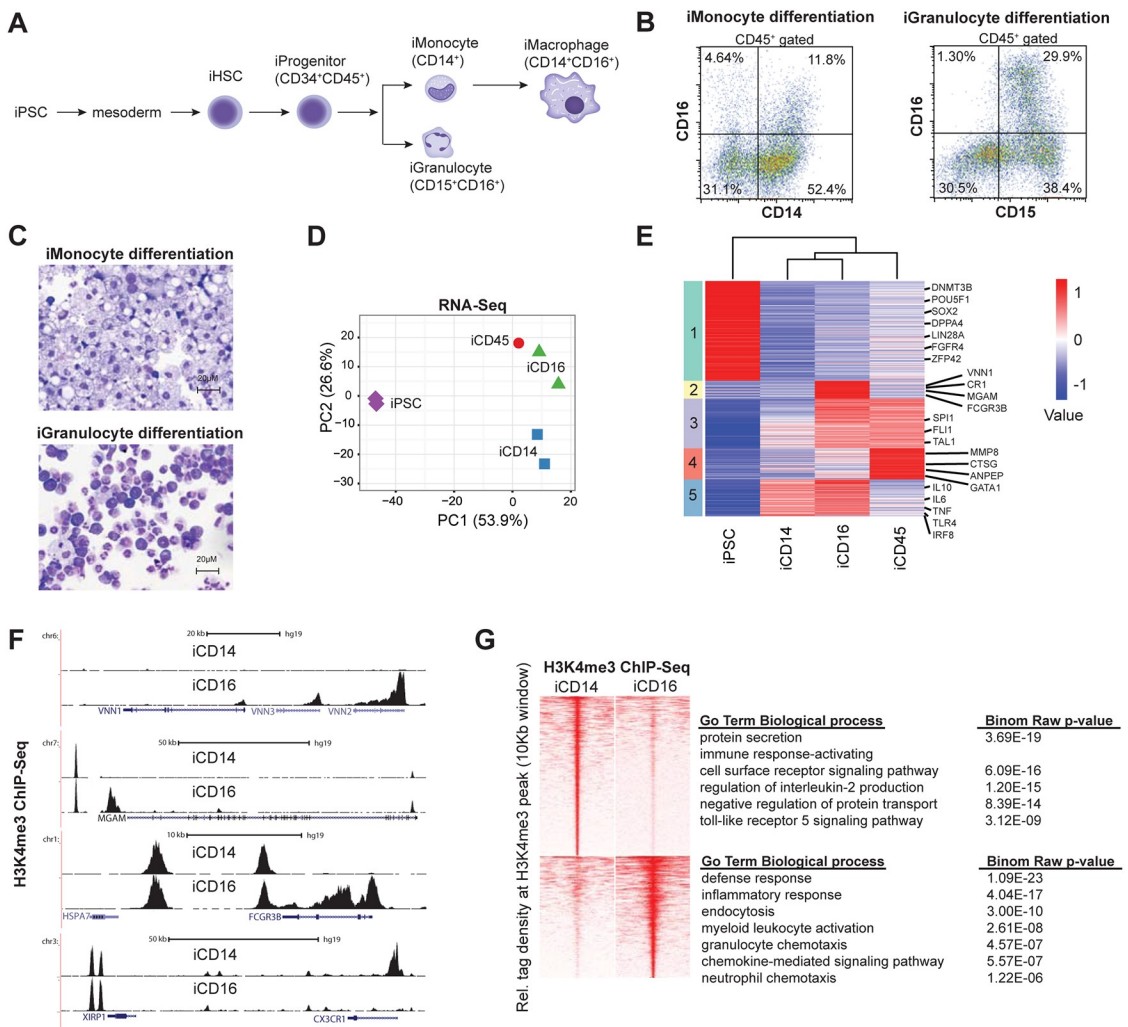

**Fig 1. Differentiation of iPSCs towards the monocytic and granulocytic lineage.** (A) Schematic representation of hematopoietic differentiation of iPSCs into iMonocytes and iGranulocytes. (B) Flow cytometric analysis was performed on differentiated iPSCs that were stained with a cocktail of antibodies directed against CD45, CD14, CD16 and CD15. Cells were first gated on the CD45 leukocyte marker, followed by analysis of CD14-CD16 and CD15-CD16 to identify iMonocytes/iMacrophages and iGranulocytes, respectively. (C) Cytospin of differentiated iPSC towards the monocytic and granulocytic lineage using May-Grunwald and Giemsa staining. (D) PCA plot of CD14⁺ iMonocytes, CD16⁺ iGranulocytes, CD45⁺ progenitors and megakaryocyte derived iPSCs (iPSC) gene expression. (E) Heatmap of iPSC as well as monocyte and granulocyte differentiated associated genes using Kmeans clustering. (F) Overview of dynamic H3K4me3 ChIP-seq signal at the VNN2, FCGR3B and CX3CR1 genomic loci. (G) Heatmap showing H3K4me3 densities in iMonocytes (CD14⁺) and iGranulocytes (CD16⁺) and the corresponding biological processes.

Finally, we used ChIP-seq with H3K4me3 antibodies to study active gene transcription and identified differential gene regulatory elements between iCD14 and iCD16 cells. We identified alterations in H3K4me3 at many differentially expressed genes, such as VNN1, FCGR3B and MGAM (Fig 1F) corroborating the expression analysis. We used MACS peak calling [24] to identify H3K4me3 occupied regions and compared iMonocytes/iMacrophages to iGranulocytes. Gene ontology analysis of differential occupied regions showed that monocyte/macrophage pathways such as immune response were associated with iCD14 peaks, while granulocyte specific pathways such as granulocyte chemotaxis were correlated with iCD16 upregulated peaks (Fig 1G).

Together, these data suggest that iPSCs can be differentiated towards the monocytic and granulocytic lineage.

### Expression of AML1-ETO inhibits granulocyte differentiation

To assess the role of AML1-ETO in leukemia development, we induced AML1-ETO expression during iPSC hematopoietic differentiation by addition of doxycycline. The dox concentration was previously experimentally determined using RT-qPCR in iPSCs which revealed expression of AML1-ETO at a dox concentration of 14 ng/ml (Fig 2A) mimics levels in the model cell line Kasumi-1 [18]. By western blot analysis we confirmed that AML1-ETO protein levels were induced in dox-treated cells (Fig 2B). Next, we examined the effects of AML1-ETO expression on iPSC differentiation by FACS analysis. Induction of AML1-ETO with dox at day 10 followed by FACS analysis after granulocyte differentiation revealed a 3-fold reduction (from 18.5% to 6.4%) of CD15$^+$CD16$^+$ cells (Fig 2C, bottom panels). The reduction of CD15$^+$CD16$^+$ cells coincided with an increase in CD34$^+$CD45$^+$ progenitor cells from 9.1% to 25.2% (Fig 2C, top panels). Decreased CD15$^+$CD16$^+$ and increased CD34$^+$CD45$^+$ cell numbers were also observed at other time points of dox induction (for example dox induction at day 6 or at day 14) during granulocytic differentiation (see representative experiment in Fig 2D and S2A Fig) and could be validated and statistically confirmed in independent replicate experiments (S2B Fig). In contrast, we observed only marginal effects of AML1-ETO expression on

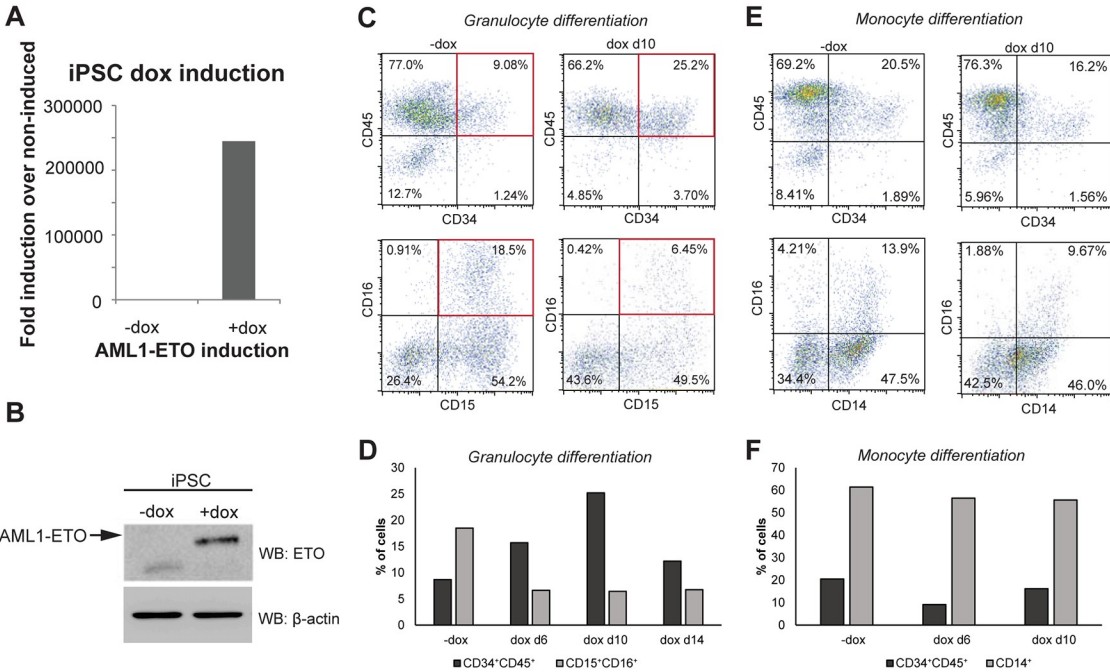

**Fig 2. Induction of AML1-ETO leads to increased CD34$^+$CD45$^+$ granulocytic progenitors.** (A) RT-qPCR analysis of iPSC before and after induction of AML1-ETO with 14ng/ml dox for 48 hours using primers located at the AML1-ETO fusion point. Data are normalized to the housekeeping gene GAPDH. (B) Western blot analysis identified AML1-ETO expression in iPSCs after stimulation with 14 ng/ml dox for 48 hours using an ETO specific antibody. B-actin was used as loading control. (C-F) Flow cytometry analysis of control and AML1-ETO differentiated iPSCs towards the monocytic (E-F) and granulocytic (C-D) lineage. Progenitor cells were characterized using CD34 and CD45 antibodies. To identify the iMonocytes/iMacrohages and iGranulocytes, cells were first gated on CD45 followed by analysis of CD14, CD16 and CD15 markers. (D, F) Bar diagram showing the percentage of CD34$^+$CD45$^+$ (progenitors), CD14$^+$ (iMonocytes/iMacrophages) and CD15$^+$CD16$^+$ (iGranulocytes) cells in AML1-ETO expressing (+dox) and control cells (-dox) for one representative experiment. Results of replicate experiments are shown in the S2B Fig.

monocytic differentiation (Fig 2E and 2F, S2A and S2B Fig), with limited changes in the percentages of CD34$^+$CD45$^+$ and CD14$^+$ cells.

Together, these results corroborate previous findings using AML1-ETO transduced progenitors [8, 9] and suggest that AML1-ETO during iPSC differentiation targets specifically granulocytic development and alters this pathway most likely at a progenitor stage of this lineage.

### Time-course RNA-seq analysis of AML1-ETO expressing iPSC differentiation

To analyze the effect of AML1-ETO on the different stages of hematopoietic differentiation, we harvested dox-treated and control cells at 8 stages (either adherent or suspension) of *in vitro* granulocyte differentiation (S3A Fig) and performed bulk RNA-seq for each time point. AML1-ETO was induced from the start of differentiation and its expression was maintained by continuous addition of dox and independent RNA-seq replicates of the dox induced samples showed good reproducibility (S3B Fig). Transcriptome analysis and Kmeans clustering of the RNA-seq profiles revealed two main clusters, one represented by RNA extracted from adherent cells and one by RNA from suspension cells (Fig 3A), while the expression of AML1-ETO did not significantly affect this clustering. In our time course experiment we detected low expression of RUNX1 at the early stages of adherent cell differentiation whereas increased levels were observed in the late adherent and suspension cells (S4A Fig), consistent with a role for RUNX1 in hematopoietic differentiation. The expression of RUNX1 from day 20 onwards suggests effects of AML1-ETO will be most prominent after day 20, i.e. after activation of the RUNX1 gene program.

Gene Ontology pathway analysis was performed to identify gene shifts during the adherent (endothelial) to suspension (hematopoietic) transition (EHT) (Fig 3A, right, S1 Table), specifically focusing on the two major clusters 2 (yellow) and 5 (blue). Genes that were upregulated in the adherent cell fraction (cluster 5) included BMPs, Cadherins and Collagens and were associated with "developmental process", "cell migration" and "cell adhesion" terms. In contrast, the suspension cells (cluster 2) showed increased expression of genes such as MPO, FCGR3B, SPI1 and CD33 that are significantly enriched in "immune system process", myeloid leukocyte activation" and inflammatory response" pathways. PCA analysis confirmed our observation and showed separation of the adherent and suspension cells by PC1 whereas PC2 separates the early (day 0–10) and late differentiated (day 20 and 30) adherent cells (Fig 3B).

### AML1-ETO inhibits myeloid cell differentiation and activates blood coagulation pathways in iPSC derived granulocytes

To assess the transcriptional changes induced by AML1-ETO upon activation of the RUNX1 gene program, we compared RNA-Seq data of control and dox treated granulocytic cells at day 31 and day 40 using Kmeans cluster analysis, revealing 5 clusters (Fig 4A). Expression of AML1-ETO leads to a downregulation of genes such as STAT1, SPI1 and RXRA involved in myeloid cell differentiation (cluster 3, S2 Table) which is consistent with our cell surface marker analysis showing that AML1-ETO inhibits granulocyte differentiation. Interestingly, at day 31 of granulocyte differentiation we observed that AML1-ETO stimulates the expression of genes involved in metabolism (cluster 2), while these genes are again downregulated at day 40 of granulocyte differentiation, suggesting that changes in metabolism might be required for early transformation to affect other critical cellular processes or that cells have become quiescent [25]. Further analysis revealed that AML1-ETO increases expression of genes that regulate blood coagulation and angiogenesis (cluster 4) such as PDGFA, PDGFB and several platelet

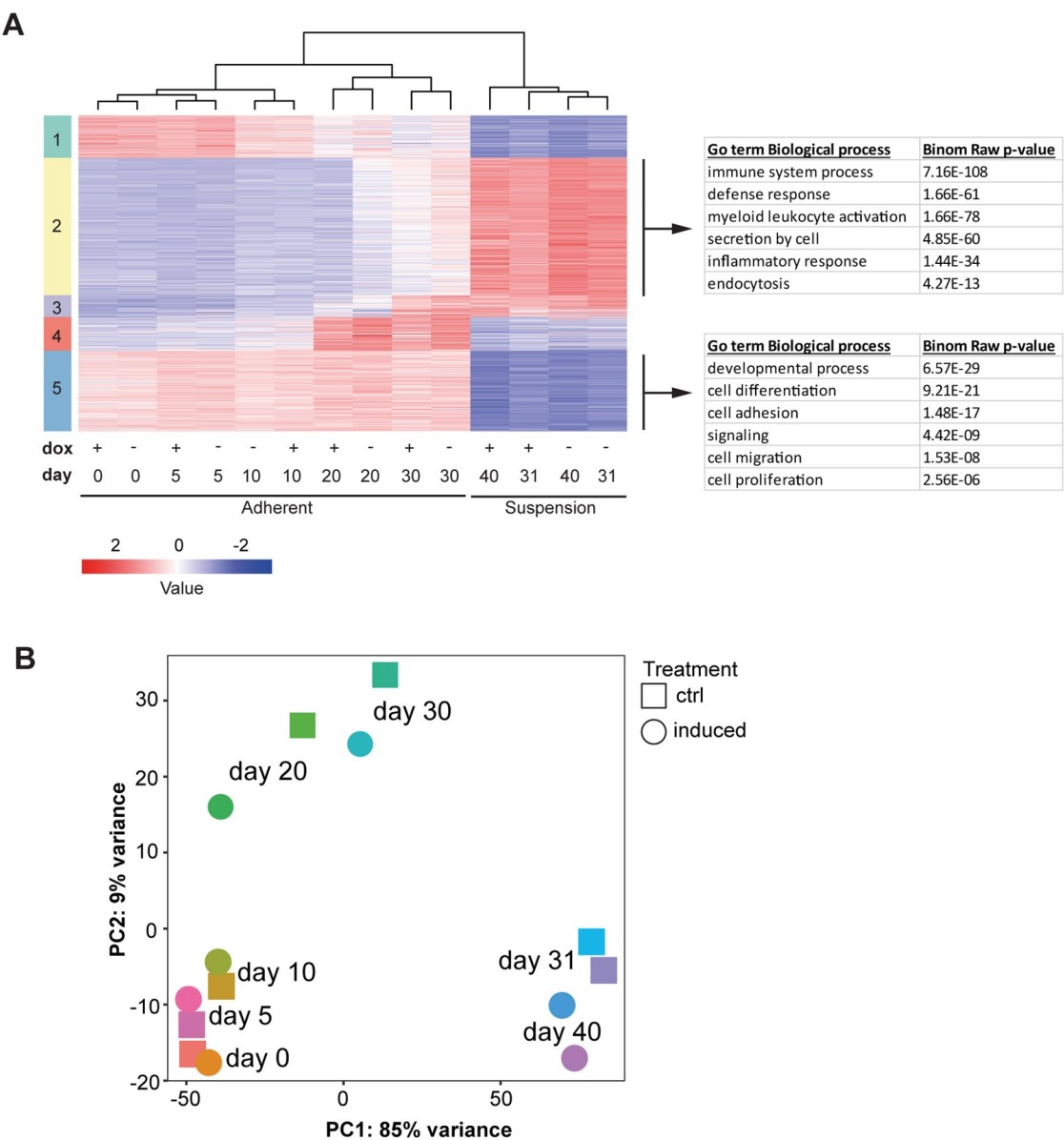

**Fig 3. RNA-seq analysis of AML1-ETO iPSCs during granulocyte differentiation.** (A) Heatmap of differential genes from control (-dox) and AML1-ETO (+dox) iPSCs differentiated towards the granulocytic pathway using K-means clustering. RNA samples were collected at different days during granulocyte differentiation. The adherent and suspension clusters were further identified by Gene ontology biological pathway analysis. (B) PCA plot of control (-dox) and AML1-ETO (+dox, induced) RNA-seq samples collected at various time points during granulocyte differentiation.

glycoproteins, processes that are frequently targeted in leukemia to promote proliferation of blast cells and to increase bone marrow vasculature for survival of leukemia [26–29].

To validate our findings we selected several genes deregulated in day 40 differentiated AML1-ETO expressing iPSCs and tested expression in independent differentiation experiments using RT-qPCR. The result revealed consistent patterns of up- and downregulated genes (S4B Fig).

To probe the difference in the AML1-ETO gene program between AML1-ETO iPSCs (one oncogenic hit cells) and cell lines (multi oncogenic hit cells), we compared expression data of

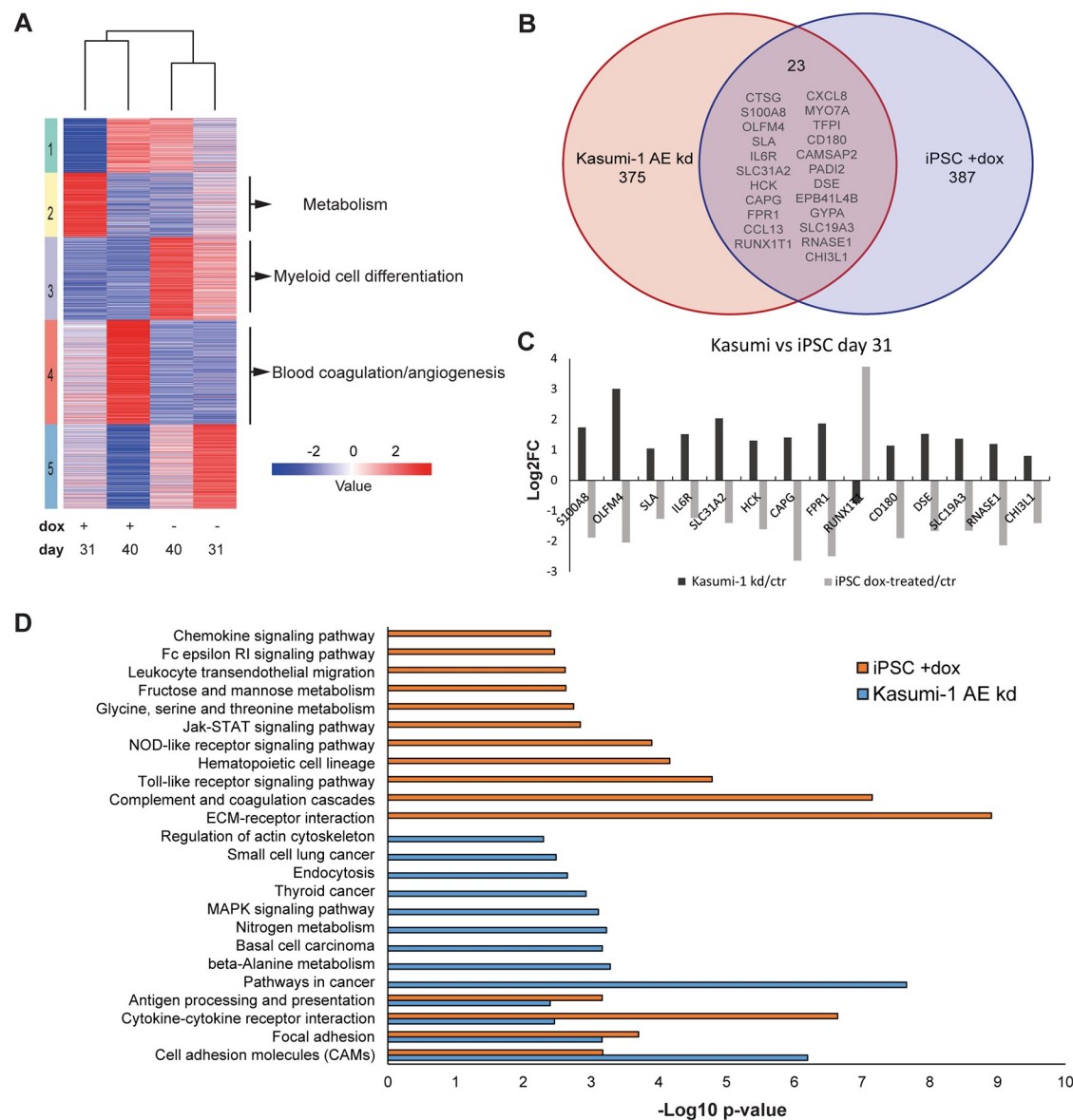

**Fig 4. AML1-ETO affects critical cellular processes during granulocytic differentiation.** (A) Heatmap of control (-dox) and AML1-ETO (+dox) granulocytic associated genes using Kmeans clustering. RNA samples were collected at the suspension cell stage (d31 and d40). The specific biological processes were identified for each cluster using Gene Ontology pathway analysis. (B) Venn diagram comparing gene expression of Kasumi-1 AML1-ETO knockdown cells (398 genes) with iPSC induced AML1-ETO cells (410 genes), of which 23 genes were identified in both datasets. (C) Bardiagram showing the log 2-fold change of genes that were regulated in opposing direction in Kasumi-1 AML1-ETO knockdown cells compared to AML1-ETO induced iPSCs. (D) Identification of unique and overlapping KEGG pathways in Kasumi-1 AML1-ETO knockdown cells as well as AML1-ETO induced iPSCs using Gene Set Enrichment Analysis. The bardiagram displays the -Log10 p-value of the pathways.

AML1-ETO expressing iPSCs to Kasumi-1 in which AML1-ETO was knocked down [30] by selecting genes that show a log2-fold change (up or down) between control and knockdown AML1-ETO (AE) cells. We found that there were 23 genes affected both by AML1-ETO induced expression in iPSCs and by AML1-ETO knockdown in Kasumi-1, while 387 genes (S3 Table) showed specific altered expression in AML1-ETO induced iPSCs (Fig 4B), further extending the repertoire of putative AML1-ETO target genes reported in literature [18, 30, 31]

and suggesting additional oncogenic hits are essential to make cells carrying one driver mutation leukemic. Interestingly, while the majority of changed genes showed reduced expression, in line with AML1-ETO acting as a transcriptional repressor, 30% of changed genes showed increased expression, revealing that AML1-ETO might also be involved in transcriptional activation. The genes that are common between the Kasumi-1 AML1-ETO knockdown and the AML1-ETO induced iPSC showed opposite expression levels, i.e. upon expression of AML1-ETO in iPSCs gene expression is reduced, while upon removal of AML1-ETO in Kasumi-1 gene expression is increased (Fig 4C). Together, these results identify a core set of genes repressed by AML1-ETO during leukemogenesis.

Next, all 410 genes (23 common and 387 iPSC specific) that were altered upon AML1-ETO induction in iPSC were further analyzed by Gene Set Enrichment Analysis to define specific pathways that are targeted. Consistent with the literature [18, 32, 33], we observed that AML1-ETO affected signaling, metabolic and hematopoietic specific pathways, processes that are often found to be deregulated in AML [34–36]. Similar analysis was performed with the 375 unique genes of Kasumi AE knockdown cells and revealed enrichment for pathways in cancer and metabolism, corresponding with its mature phenotype. This analysis revealed only 4 pathways, related to cell adhesion and immune regulation, that were deregulated both in the AML1-ETO expressing iPSC and in the Kasumi-1 cells upon AML1-ETO knockdown (Fig 4D), suggesting these are core to AML1-ETOs oncogenic activity.

Overall, these analyses uncovered additional putative AML1-ETO target genes and revealed that the aberrant AML1-ETO gene program affects many important biological processes, but particularly cell adhesion and immune activation.

## AML1-ETO expression is associated with altered H3K27ac patterns

AML1-ETO is thought to affect gene expression by recruiting epigenetic regulator complexes, in particular those containing HDAC/HAT activity, to target genes [13]. To identify genes that potentially represent targets of the AML1-ETO epigenetic regulation machinery, we performed H3K27ac ChIP-Seq in dox induced and control iPS cells during differentiation towards the granulocytic lineage. H3K27ac is associated with gene activity, marking both promoters and enhancers and behaves dynamic in various cellular differentiation systems. Using deeptools analysis [37] we identified 1,080 regions 2-fold changed in acetylation such as at the WT1 and F3 genomic loci (Fig 5A). Consistent with the reported HDAC recruitment of AML1-ETO [13], we observed reduced H3K27ac signal in the majority of dynamic regions upon induction of AML1-ETO (Fig 5B). These regions include well defined target genes such as SPI1 and CEBPA. Only 353 regions showed enrichment of acetylation.

Consistent with previous findings, functional analysis of genes associated with dynamic H3K27ac peaks showed involvement in many signaling pathways, metabolism and transendothelial migration (Fig 5C) as has been reported for AML1-ETO binding sites before [31, 38]. Similarly, as for AML1-ETO binding [18], annotation of the dynamic peaks revealed that 50% of dynamic peaks are located in transcription start sites (TSSs), while the other half is located in putative enhancer regions (introns and intergenic regions) (Fig 5D).

To examine the correlation between the transcriptome and the H3K27ac ChIP-seq dataset, we generated a scatter plot calculating the LFC of differentially expressed genes (2-fold difference) and the H3K27ac peaks that are located within 2kb. This showed a positive correlation (p< 2.20E-16) between the RNA- and ChIP-Seq datasets (Fig 5E), suggesting that, as expected, reduced H3K27ac correlates with reduced expression levels.

To examine putative binding of AML1-ETO and other transcription factors (TFs) at the dynamic H3K27ac regions, we performed motif analysis using HOMER. This analysis revealed

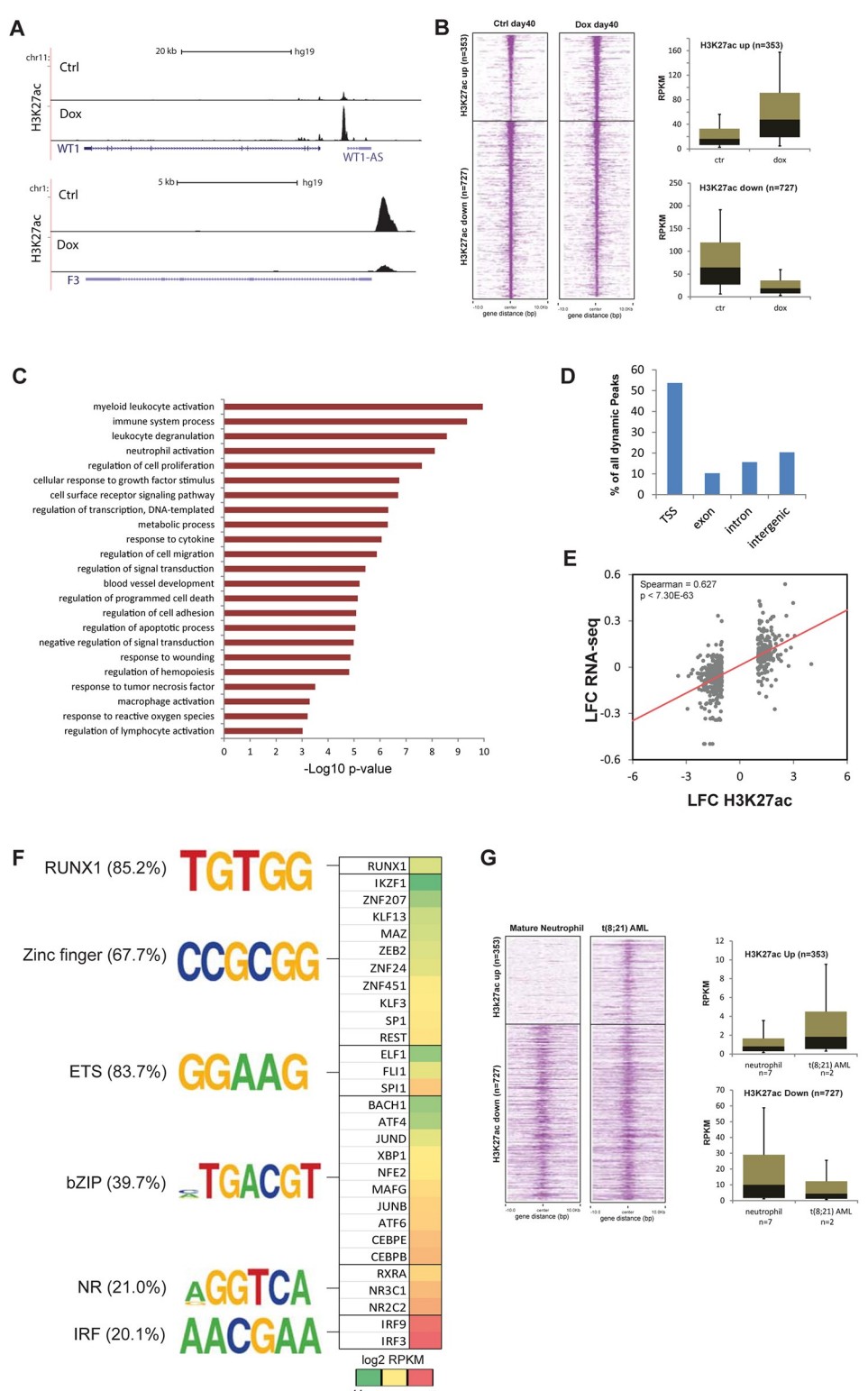

**Fig 5. AML1-ETO downregulated genes are hypoacetylated.** (A) Overview of dynamic H3K27ac ChIP-seq signal at the WT1 and F3 genomic loci. (B) Left. Density heatmap showing up and downregulated H3K27ac peak regions in control and AML1-ETO induced iPSCs. Right. Boxplot showing the RPKM values, computed from H3K27ac, in control and AML1-ETO induced iPSCs. (C) GO analysis of all dynamic H3K27ac peaks in AML1-ETO induced iPSCs. The bardiagram displays the–Log10 p-value of the pathways. (D) Genomic distribution of all dynamic H3K27ac peaks

divided in TSS (transcription start site), exon, intron and intergenic. (E) Scatter plot comparing the log fold change of H3K27ac peaks versus the log fold change of gene expression using spearman correlation. (F) Overview of the AML1-ETO binding motifs in all H3K27ac dynamic peaks. (G) Left. Density heatmap of H3K27ac signal observed in mature neutrophils and primary t(8;21) AMLs at up and downregulated H3K27ac peak regions defined in control and AML1-ETO induced iPSCs in panel B. Right. Boxplot showing the RPKM values computed from H3K27ac in 7 mature neutrophil and 2 primary t(8;21) AML samples at the regions defined in panel B.

enrichment of the RUNX consensus motif (Fig 5F), confirming a role for AML1-ETO in deregulating H3K27ac dynamics at these sites. In addition, consensus motifs for ETS, Zinc finger, bZIP, IRF and nuclear receptors were found (Fig 5F). Expression analysis of all TFs [39] expressed in AML1-ETO iPSCs identified several high expressed transcription factors belonging to these families as likely candidates of AML1-ETO coregulators including RUNX1, IKZF1, SPI1 and BACH1 (Fig 5F).

Finally, to investigate whether the dynamic H3K27ac regions between AML1-ETO expressing and normal differentiating iGranulocytes reflect differences observed between primary granulocytes and AML1-ETO expressing AMLs we examined the same H3K27ac increased and decreased peak regions in 7 mature neutrophils (http://dcc.blueprint-epigenome.eu/#/datasets/EGAD00001000930) and two t(8;21) positive primary AML patient samples (http://dcc.blueprint-epigenome.eu/#/datasets/EGAD00001001481; S00Y4YH1 and S013SSH1). The H3K27ac density (RPKM) at these peaks was computed and the average was calculated for all neutrophils and AML samples. This revealed that neutrophils have increased H3K27ac signal compared to AML1-ETO positive AML samples in regions that go down in H3K27ac after AML1-ETO expression during iPSC differentiation (Fig 5G). Similarly, regions that increase in H3K27ac ('Up' regions) are enriched in t(8;21) AMLs compared to neutrophils, suggesting the AML1-ETO expression during in vitro iPSC differentiation recapitulates changes observed in primary cells, and hence, can serve as an additional model to study AML1-ETO pathogenesis.

Together, these results suggest that potentially through deregulation of HAT/HDAC recruitment, AML1-ETO could be involved in altering the gene program of iPS cells differentiating towards granulocytes to induce t(8;21) leukemic characteristics.

## Discussion

The molecular pathways targeted by the AML1-ETO oncofusion protein are very well studied in mouse models, AML patient blasts and cell lines [18, 30, 31, 38, 40, 41]. However, these studies did not provide full insight into the single effect of AML1-ETO on leukemic transformation in human, nor the early effects of AML1-ETO at pre-leukemic stages. Although transduction of hematopoietic progenitor cells has been used for this purpose [14, 38, 49] the variability in the source and quality of the cells used for transduction impairs reproducibility and comparison of (NGS) datasets generated using these systems. Therefore, we and others (https://www.biorxiv.org/content/biorxiv/early/2019/08/28/748921.full.pdf) generated an inducible human iPSC model and expressed the AML1-ETO oncofusion protein at various endothelial and hematopoietic differentiation stages. Our system allowed to perform *in vitro* AML1-ETO deregulated transcriptome and epigenome analysis and identify deregulated genes and pathways in this otherwise non-mutated background. Moreover, it provides a stable and reproducible system for examining and comparing different oncogenes on the same genetic background.

As AML1-ETO is associated with myeloid leukemia and has been shown to inhibit granulocytic differentiation [9, 42, 43], we examined its effect on myeloid development in our iPSC model by adding dox at various time points during hematopoietic differentiation. In line with

these findings, we confirmed that AML1-ETO affects specifically the granulocytic lineage leading to reduced numbers of CD15$^+$CD16$^+$ granulocytes and simultaneously increased CD34$^+$CD45$^+$ progenitors. In contrast to the effect on granulocyte differentiation, AML1-ETO did not affect monocyte differentiation, indicating that it specifically targets the granulocytic progenitors in iPSCs differentiation. The absence of an effect on monocyte differentiation might be explained by the fact that iPSCs show embryonic natures and are likely to be differentiated as a mimic of primitive hematopoiesis cells [44]. This is in line with the finding that the t (8;21) translocation can also give rise to childhood leukemia [45]. To study the effect of AML1-ETO on definitive hematopoiesis in iPSCs, iPS cells likely need to be stimulated with exogenous cytokines, feeder cells, and extracellular matrix-coated dishes [44].

Previously it has been shown that hematopoietic differentiation of iPS cells occurs via the hemogenic endothelial transition [46]. We assessed if AML1-ETO also affects early endothelial to hematopoietic development by investigating various differentiation stages and observed no effect of AML1-ETO on the adherent and suspension cell clusters. However, AML1-ETO has a moderate effect on hematopoietic genes, such as MPO, ITGA2B, GATA2 in the suspension cells, while endothelial genes, such as PECAM1, KDR and FZD4 were normally expressed in dox induced adherent cells. This confirms our findings, as well as previous, that AML1-ETO specifically affects the development of granulocytes [9].

The Kasumi-1 cell line harboring the t(8;21) translocation is frequently used to assess the function of AML1-ETO and identify its molecular interactors [18, 30]. To identify new targets of AML1-ETO we compared the AML1-ETO expressing iGranulocytes to the Kasumi-1 AE knockdown cells showing 23 genes overlapping, while 387 unique genes were found in the dox induced iPSCs. All overlapping genes appeared to be downregulated by AML1-ETO, which is consistent with its repressor function [2, 14]. Still, of the remaining 387 changed genes in iPSCs, 293 went down and 94 went up, corroborating previous observations that AML1-ETO can both activate as well as repress genes [13, 47, 48].

AML1-ETO is defined as a repressor by recruiting HDAC complexes [13], which corresponds to our finding that induction of AML1-ETO in iPSCs results in reduced H3K27 acetylation signal at 727 regions compared to control cells. In line with this, we showed that neutrophils have a higher acetylation signal compared to AML samples at these reduced H3K27ac regions. In contrast, the 353 regions that show increased acetylation are not enriched in neutrophils, but are enriched in primary t(8;21) AMLs.

Together our results reveal the oncogenic activities of AML1-ETO in a human cell system in the absence of additional mutations. It revealed similar self-renewal capacity of progenitor cells expressing AML1-ETO, as evidenced by increased/sustained levels of CD34+ cells, as observed by others [14, 38, 49], but a difference in proliferative capacity, potentially due to the cell of origin in which AML1-ETO was expressed. While monocyte differentiation seems hardly affected by AML1-ETO expression, in particular granulocyte differentiation is affected. Finally, It identified new putative AML1-ETO targets and key deregulated biological processes that are likely contributing to leukemogenesis.

## Supporting information

**S1 Fig. Reproducible granulocyte or monocyte differentiation of iPSC.** Flow cytometric analysis was performed on iPSCs differentiated towards the granulocyte or monocyte lineage (n = 12). Cells were stained with a cocktail of antibodies directed against CD45, CD14, CD16 and CD15. Cells were first gated on the CD45 leukocyte marker, followed by analysis of CD14-CD16 and CD15-CD16 to identify iMonocytes/iMacrophages and iGranulocytes, respectively. (PDF)

**S2 Fig. Induction of AML1-ETO leads to reduced CD15⁺CD16⁺ granulocytes. A**. Flow cytometry analysis of control and AML1-ETO differentiated iPSCs towards the granulocytic and monocytic lineage. AML1-ETO was induced at day 0, 6, 10 and 14 of differentiation using doxycycline. Progenitor cells were characterized using CD34 and CD45 antibodies. To identify the iMonocytes/iMacrohages and iGranulocytes, cells were first gated on CD45 followed by analysis of CD14, CD16 and CD15 markers. **B**. Flow cytometry analysis of replicates (n = 7) of control and AML1-ETO differentiated iPSCs towards the monocytic (right) and granulocytic (left) lineage. Progenitor cells were characterized using CD34 and CD45 antibodies. To identify the iMonocytes/iMacrohages and iGranulocytes, cells were first gated on CD45 followed by analysis of CD14, CD16 and CD15 markers. * p-value <0.05, n.s. not significant.
(PDF)

**S3 Fig. RNA-seq analysis of control and AML1-ETO induced cells during granulocytic differentiation. A**. Schematic overview of hematopoietic differentiation towards the granulocytic lineage and the samples that were collected during hematopoietic differentiation of control and AML1-ETO induced cells. Adherent cells were collected from day 0–40 whereas suspension cells were collected from day 20–40. **B**. Scatterplot showing the log transformed normalized reads of replicate RNA-seq samples of the indicated time points. The blue line represents the trendline with the correlation coefficient depicted.
(PDF)

**S4 Fig. RUNX1 expression during hematopoietic differentiation and differential gene expression validation. A**. (Left) RPKM expression of RUNX1 gene in control (-dox) and AML1-ETO (+dox) differentiated iPSC at various time points during iGranulocyte development. (Right) Replicate analysis of RPKM expression of RUNX1 gene in control (-dox) and AML1-ETO (+dox) differentiated iPSC at two time points during iGranulocyte development. **B**. RT-qPCR validation of a subset of genes differentially expressed (based on the RNA-seq analysis) in AML1-ETO expressing and control differentiated iPSCs. For both the RT-qPCR experiment and the RNA-seq experiment the resulting values for the non-induced situation (–dox) for each gene was set at 1. The relative change according to this normalization is depicted in the figure.
(PDF)

**S1 Table. Gene ontology pathway analysis belonging to Fig 3A.**
(PDF)

**S2 Table. Gene ontology pathway analysis belonging to Fig 4A.**
(PDF)

**S3 Table. Genes differentially expressed in AML1-ETO expressing iPSCs.**
(PDF)

**S1 Raw Images. AML1-ETO western blot.**
(TIF)

**S2 Raw Images. AML1-ETO western blot control.**
(TIF)

**S3 Raw Images. Cytospin iMonocyte differentiation.**
(TIF)

**S4 Raw Images. Cytospin iGranulocyte differentiation.**
(TIF)

## Author Contributions

**Conceptualization:** Esther Tijchon, Amit Mandoli.

**Data curation:** Esther Tijchon, Amit Mandoli, Jos G. A. Smits, Francesco Ferrari, Branco M. H. Heuts, Falco Wijnen, Eva M. Janssen-Megens, Joost H. A. Martens.

**Formal analysis:** Esther Tijchon, Guoqiang Yi, Amit Mandoli, Jos G. A. Smits, Francesco Ferrari, Branco M. H. Heuts, Bowon Kim, Eva M. Janssen-Megens, Jan Jacob Schuringa, Joost H. A. Martens.

**Funding acquisition:** Joost H. A. Martens.

**Investigation:** Esther Tijchon, Guoqiang Yi, Jos G. A. Smits, Francesco Ferrari, Falco Wijnen, Bowon Kim, Eva M. Janssen-Megens, Jan Jacob Schuringa.

**Methodology:** Esther Tijchon, Guoqiang Yi, Amit Mandoli.

**Supervision:** Joost H. A. Martens.

**Validation:** Esther Tijchon, Guoqiang Yi, Branco M. H. Heuts.

**Visualization:** Esther Tijchon, Amit Mandoli.

**Writing – original draft:** Esther Tijchon, Joost H. A. Martens.

**Writing – review & editing:** Joost H. A. Martens.

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
