## [Decision Letter · Decision Letter 0]

10 Oct 2019

PONE-D-19-26534

The acute myeloid leukemia associated AML1-ETO fusion protein alters the transcriptome and cellular progression in a single-oncogene expressing in vitro induced pluripotent stem cell based granulocyte differentiation model

PLOS ONE

Dear Dr Martens,

Thank you for submitting your manuscript to PLOS ONE. After careful consideration, we feel that it has merit but does not fully meet PLOS ONE’s publication criteria as it currently stands. Therefore, we invite you to submit a revised version of the manuscript that addresses the points raised during the review process. In particular, as pointed out by the reviewer 2, some data will need appropriate statistical analyses. 

We would appreciate receiving your revised manuscript by Nov 24 2019 11:59PM. To enhance the reproducibility of your results, we recommend that if applicable you deposit your laboratory protocols in protocols.io, where a protocol can be assigned its own identifier (DOI) such that it can be cited independently in the future. For instructions see: http://journals.plos.org/plosone/s/submission-guidelines#loc-laboratory-protocols

We look forward to receiving your revised manuscript.

Kind regards,

Arun Rishi, Ph.D.

Academic Editor

PLOS ONE

Journal Requirements:

1. Please note that all PLOS journals ask authors to adhere to our policies for sharing of data and materials: https://journals.plos.org/plosone/s/data-availability. According to PLOS ONE’s Data Availability policy, we require that the minimal dataset underlying results reported in the submission must be made immediately and freely available at the time of publication. As such, please remove any instances of 'unpublished data' or 'data not shown' in your manuscript and replace these with either the relevant data (in the form of additional figures, tables or descriptive text, as appropriate), a citation to where the data can be found, or remove altogether any statements supported by data not presented in the manuscript. If you decide to include the animal research data in your manuscript, please also include a section in the Methods regarding animal welfare and related experiments (the number of animals used in your study, the frequency of animal monitoring, the methods of anesthesia, analgesia, euthanasia as well as the approval number from your institutional animal ethics committee.

2. In your Methods section, please include culturing details and provide the source of the Kasumi-1 cell line used in your study.

Reviewers' comments:

Reviewer's Responses to Questions

**Comments to the Author**

1. Is the manuscript technically sound, and do the data support the conclusions?

Reviewer #1: Yes

Reviewer #2: Yes

2. Has the statistical analysis been performed appropriately and rigorously? 

Reviewer #1: Yes

Reviewer #2: No

3. Have the authors made all data underlying the findings in their manuscript fully available?

Reviewer #1: Yes

Reviewer #2: Yes

4. Is the manuscript presented in an intelligible fashion and written in standard English?

Reviewer #1: Yes

Reviewer #2: Yes

5. Review Comments to the Author

Reviewer #1: Tijchon and colleagues examine the effect of AML-1ETO9a on the Differentiation of IPSC. They conclusively demonstrated how AML1-ETO9 influences epigenetic changes.

However they should discuss how their manuscript differs from other publications in which AML1-ETO9a was expressed inc CD 34 positive cells and what is specifically the really new finding.

With regard to the experiments, they have been conducted in an excellent manner.

Reviewer #2: This is an interesting study on the function of the fusion protein AML1-ETO that is produced in t(8:21) AML. The authors use iPS cells that can be reprogrammed into myeloid and granulocytic lineages to assess the effect of a spontaneous expression of an AML1-ETO fusion protein in a non-malignant stem cell background. This experimental system offers the possibility to investigate AML1-ETO functions at a molecular level before other, secondary events take place. A number of observations can be made:

The MACS sort is based on one single marker either CD16 or CD14. It seems that this sort may contaminate the CD14+ population with CD16 cells under conditions of monocyte differentiation. The authors should elaborate on this in the text whether this is a concern or not.

Fig. 1: What is the rationale to look specifically at H3K4me3?

Fig. 2D and E: The data shown in the bar graph does not contain any statistical parameters (errors, p values). How much redundancy is there between Figure 2 D/F and Fig S 2B? Is the difference in Fig. S2 B for monocytes significant?

Line 289: How were the stages determined at which the cells were harvested for RNA-Seq?

Fig. S4A: Similar to Fig. 2D1E, this Figure does not contain any statistical parameters (errors, p values). Does the PCR detect only the endogenous RUNX1 and the part in the RUNX1-ETO fusion gene? Is this result expected?

Fig. 3B: The way the PCA is depicted makes it difficult to understand the point. Maybe there is a better way to represent these data - possibly a separation for each day of treatment?

Fig. S4B: This experiment should be better explained; in particular how the baselines for the PCR and the RPKM were set.

Line 371: Why was this specific histone mark chosen and not for instance H3K9?

Lines 397-409: This experiment is less well explained compared to the others and the significance of the finding remains somewhat unclear. Also, to use the term acetylome if only one acetylated residue is measured is likely going a bit far. I suggest to avoid this term here. The conclusion that si put at the end of this section is far reaching and should be moved to the discussion and even there should be tuned down.

Minor comments:

Line 241: it would be more appropriate to mention “gene signatures” instead of “gene patterns”

6. PLOS authors have the option to publish the peer review history of their article (what does this mean?). If published, this will include your full peer review and any attached files.

Reviewer #1: No

Reviewer #2: No

---

## [Author Response · Author response to Decision Letter 0]

8 Nov 2019

Dear Dr. Rishi,

Thank you and the reviewers for the helpful comments and suggestions to our manuscript, “The acute myeloid leukemia associated AML1-ETO fusion protein alters the transcriptome and cellular progression in a single-oncogene expressing in vitro induced pluripotent stem cell based granulocyte differentiation model” (PONE-D-19-25306). Based on these comments, we performed several new analysis including statistical tests to further strengthen our claims. In addition, upon request by the reviewers, we added further clarifications to some of our experiments and interpretation. The point-by-point responses to the referees’ comments are listed below. All amendments are highlighted in yellow in the ‘track changes’ version of the manuscript, raw image files are in the supporting information and sequencing data sets have been deposited to GEO. We believe that we have carefully addressed all comments and would like to submit this revised manuscript to PLOS ONE.

We look forward to your response.

Sincerely,

Joost Martens, Ph.D.

 

Comments to the Author

Reviewer #1: Tijchon and colleagues examine the effect of AML-1ETO9a on the Differentiation of IPSC. They conclusively demonstrated how AML1-ETO9 influences epigenetic changes. However they should discuss how their manuscript differs from other publications in which AML1-ETO9a was expressed inc CD34 positive cells and what is specifically the really new finding. With regard to the experiments, they have been conducted in an excellent manner.

> We thank the reviewer for the positive evaluation of our manuscript. We referred to several manuscripts using CD34+ transduction with AML1-ETO/AML1-ETO9a in their experimental setup, but we agree with the reviewer that we could have better contrasted the findings of these studies with ours. While these studies revealed similar self-renewal capacity of progenitor cells expressing AML1-ETO, as evidenced by increased/sustained levels of CD34+ cells, a difference is observed in proliferative capacity, which is not readily seen in our system. This could be due to the cell of origin in which AML1-ETO was expressed, with CD34+ HSCs representing adult stem cells, while the iPSCs are more related to primitive hematopoietic cells. Another new finding of our study is the clear difference in lineage blocking capacity observed. While monocyte differentiation seems hardly affected by AML1-ETO expression, in particular granulocyte differentiation is affected. Finally, based on the gene expression analysis/comparison also a new set of putative target genes was identified. 

We now included the comparison with CD34+ transduction in the discussion section of our manuscript, and also added an additional reference for this. 

Reviewer #2: This is an interesting study on the function of the fusion protein AML1-ETO that is produced in t(8:21) AML. The authors use iPS cells that can be reprogrammed into myeloid and granulocytic lineages to assess the effect of a spontaneous expression of an AML1-ETO fusion protein in a non-malignant stem cell background. This experimental system offers the possibility to investigate AML1-ETO functions at a molecular level before other, secondary events take place. 

> We like to thank reviewer 2 for the positive assessment of our manuscript and the suggestions for clarification and better representation. 

The MACS sort is based on one single marker either CD16 or CD14. It seems that this sort may contaminate the CD14+ population with CD16 cells under conditions of monocyte differentiation. The authors should elaborate on this in the text whether this is a concern or not.

> We agree with the reviewer that using a single marker for monocytes also results in selection of cells carrying CD16, as is observed in Figure 1B. However, we assume that these cells still represent monocytes as apart from the classical monocyte (with high level expression of the CD14 cell surface receptor and low CD16) also non-classical monocytes exist that carry CD14 and CD16 (Ziegler-Heitbrock, L (2007). The CD14+ CD16+ Blood Monocytes: their Role in Infection and Inflammation. Journal of Leukocyte Biology). As such, we think the CD14 marker identifies the full spectrum of monocytes. This has now been included in the text on page 10. 

Fig. 1: What is the rationale to look specifically at H3K4me3?

> H3K4me3 is a stable epigenetic mark associated either with genes that are active (H3K4me3/H3K27ac) or inactive (H3K4me3/H3K27me3), and patterns have been indicative of cell type (Kellis et al., Integrative analysis of 111 reference human epigenomes, Nature). Although other marks have similar properties we selected H3K4me3, as the available antibodies are amongst the most specific and best characterized. 

Fig. 2D and E: The data shown in the bar graph does not contain any statistical parameters (errors, p values). How much redundancy is there between Figure 2 D/F and Fig S 2B? Is the difference in Fig. S2 B for monocytes significant?

> We thank the reviewer for this comment and like to clarify. The results represented in figure 2D and 2E are results of a single representative differentiation experiment and hence do not contain statistical parameters. This is now better indicated in the text. The experiment however has been repeated multiple times and these results are presented in Supplemental Figure S2B. For these we indeed did not indicate whether results were significantly different and statistical analysis has now been included. It indicates that differences upon AML1-ETO induction on granulocyte differentiation are significant, but not for the monocytes.

Line 289: How were the stages determined at which the cells were harvested for RNA-Seq?

> Given there is some variation within each individual differentiation assay, the harvesting times were set at given time points not at particular stages. The best defined stages are the transition from mesoderm differentiation to hematopoietic differentiation, the transition from adherent cells to suspension cells and the final differentiation. These are indicated in Supplemental Figure S3A. 

Fig. S4A: Similar to Fig. 2D/E, this Figure does not contain any statistical parameters (errors, p values). Does the PCR detect only the endogenous RUNX1 and the part in the RUNX1-ETO fusion gene? Is this result expected?

> Figure S4A is showing RNA-seq based RUNX1 expression during in vitro differentiation. As such it allowed us to unambiguously assign reads to RUNX1, given the uniqueness of the C-terminus (which is not present in the AML1-ETO fusion). As this is RNA-seq data and not PCR, only single RPKM values for each timepoint within the entire time course experiment are shown. We performed replicate experiments for the ‘suspension cell’ time points and now included a separate graph to indicate the variation (Supplemental Figure S4A, right). The results we obtain are in line with other iPSC based differentiation studies (Gerritsen et al., Blood Advances, 2019; Nafria et al., https://www.biorxiv.org/content/biorxiv/early/2019/08/28/748921.full.pdf) and the association of RUNX1 expression with the endothelial-to-mesenchymal transition. 

Fig. 3B: The way the PCA is depicted makes it difficult to understand the point. Maybe there is a better way to represent these data - possibly a separation for each day of treatment?

> We thank the reviewer for this comment and tried some variations for depicting the PCAn (see word file 'point-by-point' ). However, we did not feel these other representations were superior to the one we already used so settled for the one that was already included. Nevertheless, we realized it was somewhat dense, so we decided to enlarge the figure for more optimal viewing.

Fig. S4B: This experiment should be better explained; in particular how the baselines for the PCR and the RPKM were set.

> For both the RT-qPCR experiment and the RNA-seq experiment the resulting values for the non-induced situation (–dox) for each gene was set at 1. The relative change according to this normalization is depicted in the figure. This information has now been added to the figure legend.

Line 371: Why was this specific histone mark chosen and not for instance H3K9?

> We selected this mark as it is associated with activity and marks both promoters and enhancers. Moreover, it is generally conceived as very dynamic in cellular differentiation systems. Finally, as AML1-ETO mechanistically has been linked to HDAC/HAT recruitment, selecting a histone acetylation mark seemed more appropriate. This information has been added to the manuscript (page 15). 

As H3K9ac and H3K27ac patterns largely overlap (Wang et al., Nat. Genet., Combinatorial patterns of histone acetylations and methylations in the human genome), we settled for the mark for which we have most experience experimentally and for which relevant data from primary material was already available, allowing comparative analysis. 

Lines 397-409: This experiment is less well explained compared to the others and the significance of the finding remains somewhat unclear. Also, to use the term acetylome if only one acetylated residue is measured is likely going a bit far. I suggest to avoid this term here. The conclusion that is put at the end of this section is far reaching and should be moved to the discussion and even there should be tuned down.

> Although H3K27ac seems to be representative and highly correlating with several other histone acetylation marks (suggesting we are looking at the acetylome) we agree with the reviewer that based on our own results we can only comment on H3K27ac. Hence, we removed the term acetylome from the manuscript and only refer to the mark we actually examine (H3K27ac). We also rephrased the explanation regarding the experiment the reviewer refers to and tuned down the final conclusion. 

Minor comments:

> Line 241: it would be more appropriate to mention “gene signatures” instead of “gene patterns”

We changes ‘gene patterns’ into ‘gene signatures’.

 Additional questions from editorial office

1. Please note that all PLOS journals ask authors to adhere to our policies for sharing of data and materials: https://journals.plos.org/plosone/s/data-availability. According to PLOS ONE’s Data Availability policy, we require that the minimal dataset underlying results reported in the submission must be made immediately and freely available at the time of publication. As such, please remove any instances of 'unpublished data' or 'data not shown' in your manuscript and replace these with either the relevant data (in the form of additional figures, tables or descriptive text, as appropriate), a citation to where the data can be found, or remove altogether any statements supported by data not presented in the manuscript. 

> We removed the instances of unpublished data/data not shown from our manuscript. Sequence data has already been deposited to GEO and can be made directly available at the time of publication. 

2. In your Methods section, please include culturing details and provide the source of the Kasumi-1 cell line used in your study.

> This information has now been added to the Experimental procedures section. 

3. Please include captions for your Supporting Information files at the end of your manuscript, and update any in-text citations to match accordingly. 

> At the end of the manuscript we have now included an overview of the supporting information files. In addition, we uploaded the raw image files for the western blots and cytospin experiments.

> We removed the instance of unpublished data/data not shown from our manuscript and removed the phrase that referred to this data.

---

## [Decision Letter · Decision Letter 1]

27 Nov 2019

The acute myeloid leukemia associated AML1-ETO fusion protein alters the transcriptome and cellular progression in a single-oncogene expressing in vitro induced pluripotent stem cell based granulocyte differentiation model

PONE-D-19-26534R1

Dear Dr. Martens,

We are pleased to inform you that your manuscript has been judged scientifically suitable for publication and will be formally accepted for publication once it complies with all outstanding technical requirements.

With kind regards,

Arun Rishi, Ph.D.

Academic Editor

PLOS ONE

Additional Editor Comments (optional):

Reviewers' comments:

Reviewer's Responses to Questions

**Comments to the Author**

1. If the authors have adequately addressed your comments raised in a previous round of review and you feel that this manuscript is now acceptable for publication, you may indicate that here to bypass the “Comments to the Author” section, enter your conflict of interest statement in the “Confidential to Editor” section, and submit your "Accept" recommendation.

Reviewer #1: All comments have been addressed

Reviewer #2: All comments have been addressed

2. Is the manuscript technically sound, and do the data support the conclusions?

Reviewer #1: Yes

Reviewer #2: Yes

3. Has the statistical analysis been performed appropriately and rigorously? 

Reviewer #1: Yes

Reviewer #2: Yes

4. Have the authors made all data underlying the findings in their manuscript fully available?

Reviewer #1: Yes

Reviewer #2: Yes

5. Is the manuscript presented in an intelligible fashion and written in standard English?

Reviewer #1: Yes

Reviewer #2: Yes

6. Review Comments to the Author

Reviewer #1: I am happy with the modifications with no additional comments. My concerns have been Adresse. Thanks for the nice work

Reviewer #2: All my comments have been adequately addressed and I have no further concerns or questions. The manuscript provides valuable information for readers on the filed and is in its form and content suitable for publication.

7. PLOS authors have the option to publish the peer review history of their article (what does this mean?). If published, this will include your full peer review and any attached files.

Reviewer #1: No

Reviewer #2: No

---

## [Editor Report · Acceptance letter]

9 Dec 2019

PONE-D-19-26534R1 

The acute myeloid leukemia associated AML1-ETO fusion protein alters the transcriptome and cellular progression in a single-oncogene expressing in vitro induced pluripotent stem cell based granulocyte differentiation model 

Dear Dr. Martens:

I am pleased to inform you that your manuscript has been deemed suitable for publication in PLOS ONE. Congratulations! Your manuscript is now with our production department. 

With kind regards,

on behalf of

Prof Arun Rishi 

Academic Editor

PLOS ONE